# Evaluation of the Photoactivation Effect of 3% Hydrogen Peroxide in the Disinfection of Dental Implants: In Vitro Study

**DOI:** 10.3390/biomedicines11041002

**Published:** 2023-03-24

**Authors:** Ivan Katalinić, Igor Smojver, Luka Morelato, Marko Vuletić, Ana Budimir, Dragana Gabrić

**Affiliations:** 1Specialty Hospital St. Catherine, 10000 Zagreb, Croatia; ikatalinic87@gmail.com (I.K.);; 2Department of Oral Surgery, Faculty of Dental Medicine, University of Rijeka, 51000 Rijeka, Croatia; 3Department of Oral Surgery, School of Dental Medicine, University Hospital Centre Zagreb, University of Zagreb, 10000 Zagreb, Croatia; 4Department of Clinical and Molecular Microbiology, School of Medicine, University Hospital Centre Zagreb, University of Zagreb, 10000 Zagreb, Croatia

**Keywords:** photoactivation, 3% hydrogen peroxide, diode laser, periimplantitis therapy

## Abstract

Photoactivation of 3% hydrogen peroxide with a 445 nm diode laser represents a relatively new, insufficiently researched antimicrobial method in the treatment of peri-implantitis. The purpose of this work is to evaluate the effect of photoactivation of 3% hydrogen peroxide with a 445 nm diode laser, and to compare the obtained results with 0.2% chlorhexidine treatment and 3% hydrogen peroxide treatment without photoactivation, in vitro, on the surface of dental implants contaminated with *S. aureus* and *C. albicans* biofilms. Previously, 80 infected titanium implants with *S. aureus* and *C. albicans* cultures were divided into four groups: G1-negative control (no treatment), G2-positive control (0.2% chlorhexidine), G3 (3% hydrogen peroxide), and G4 (photoactivated 3% hydrogen peroxide). The number of viable microbes in each sample was determined by the colony forming unit (CFU) count. The results were statistically processed and analyzed, showing a statistically significant difference across all groups compared to the negative control (G1), and the absence of a statistically significant difference between groups G1–G3. The new antimicrobial treatment, according to the results, could be worthy of further analysis and research.

## 1. Introduction

Dental implants present a valuable therapeutic choice in the treatment of partial and complete edentulousness. However, despite the high degree of success of dental implants, vast experience and technological improvements during the last 60 years, their frequent use has also brought some very specific problems [1,2]. Peri-implant diseases (peri-implant mucositis and peri-implantitis) are probably the most significant issues associated with dental implants [3,4]. Peri-implant mucositis is defined as the presence of an inflammatory infiltrate in soft tissue induced by plaque, without loss of peri-implant bone, while peri-implantitis is a condition in which, along with soft tissue inflammation, a bone loss is present [5]. If peri-implantitis is not recognized in time and properly treated, implant loss may happen. According to Atieh et al. [6], as many as 50% of placed implants show signs of peri-mucositis, and 12–43% of implants show signs of peri-implantitis. The role of microorganisms in the development of these pathological conditions is crucial and more than 20 different species are routinely found in swabs taken from infected implants [5,7]. One specific microorganism, Staphylococcus aureus, plays a significant role in the development of peri-implantitis due to its affinity for the titanium surface of the implant and the formation of biofilms or a substrate for the growth of biofilms of other cultures, the so-called “early colonizer” [8,9]. Biofilm is defined as a microbial community of cells embedded in a polymer extracellular matrix, the creation of which is caused by microorganisms present in the matrix. Organisms inside biofilms are more resistant to different antimicrobial treatments than free, planktonic organisms [10]. Candida albicans is the most frequently isolated fungus in the human oral cavity. Despite being considered a commensal species, under certain conditions, such as periods of antibiotic use or periods of immunosuppression, it can cause mucosal infection [11]. *C. albicans* is often present in the peri-implant sulcus, both in healthy people and in patients with peri-implantitis, where it also creates a substrate for the formation of biofilms and supports inflammation [12].

The treatment of peri-implant diseases is a complex task due to the difficult access to the implant surface, limited visibility, the roughness of the titanium surface, and the pathogenicity of the microorganisms involved in the pathogenesis of these diseases [13,14,15]. Therefore, over time, numerous antimicrobial procedures and protocols have been developed. Scarano et al. [14] made an overview of the available treatments in a following manner: (1) mechanical debridement using plastic curettes, rubber polishers, ultrasonic scalers or air-powder abrasives; (2) chemical decontamination using chlorhexidine, citric acid, tetracycline, hydrogen peroxide, etc.; (3) dental laser-based treatments. Described procedures and techniques can also be mixed or combined, as in the newer study carried out by Alovisi et al. [16], where triple antibiotic paste and a glycine powder air-flow abrasion were used to fight microorganisms. However, none of the usual methods can completely remove or inactivate peri-implant pathogens due to previously mentioned factors, such as complex anatomical relationships and/or the specific implant surface [17]. Furthermore, it is even possible to damage this implant surface using certain antimicrobial treatments agents, which can then impair the healing of peri-implant tissues [18].

Although many treatment modalities exist, chlorhexidine digluconate (CHX) combined with manual debridement is still considered to be the golden standard in the treatment of periodontal and peri-implant diseases [19]. Hydrogen peroxide (H_2_O_2_) has been used in dentistry as a mouthwash to prevent plaque and as an antiseptic after oral surgery for more than 100 years [20], and it is also used in the treatment of peri-implantitis, but likely not as often as CHX. Photo-activation of H_2_O_2_, with the aim of improving the antimicrobial effect in periodontitis therapy, is a potentially interesting idea examined in the work of Mahdi et al. in 2015 [21].

They found a stronger disinfection potential when H_2_O_2_ was activated by LED light with a wavelength of 440–480 nm, compared to non-activated H_2_O_2_. Guided by this idea, the authors of this study wanted to examine the possibility of H_2_O_2_ activation with a 445 nm diode laser, and augmentation of its antimicrobial effect. According to authors knowledge, there is currently no research on this topic.

The purpose of this work is to evaluate the effect of photoactivation of 3% hydrogen peroxide with a 445 nm diode laser and compare the obtained values with chlorhexidine and hydrogen peroxide without photoactivation, in vitro, on the surface of implants contaminated with *S. aureus* and *C. albicans* biofilms.

The hypotheses of this study are:Hydrogen peroxide activated by a 445 nm diode laser shows better results in disinfection of dental implants compared to hydrogen peroxide used without 445 nm photoactivation.Hydrogen peroxide activated by a 445 nm diode laser shows equal or better results in disinfection of dental implants compared to chlorhexidine treatment.

## 2. Materials and Methods

The research was conducted at the Department of Oral Surgery of the School of dental medicine in Zagreb and at the Clinical Institute for Clinical and Molecular Microbiology of KBC Zagreb, Croatia. Eighty Zimmer Biomet Tapered Screw-Vent MTX 4.1/10 mm titanium implants (Zimmer Biomet, Palm Beach Gardens, FL, USA) were used in the study. According to Ehrenfest et al. [22], these implants are grade 5 titanium core, with a surface modified by sand-blasting (resorbable blasting media—RBM like calcium phosphate). The surface was microrough, nano-smooth and homogenous. The implants used in this study were contaminated with cultures of *S. aureus* and *C. albicans* isolated from clinical samples at University hospital centre Zagreb. Bacterial and fungal strains were grown separately on Columbia agar for 72 h, after preparation of separate bacterial and fungal suspensions using a thioglycolate broth. They were then mixed into a common suspension. An optical densitometer (Densimat, Biomerieux, Marcyl’Etoile, France) determined the density at 600 nm, which corresponded to 1 × 10^8^ CFU/mL. All dental implants were immersed in 0.3 mL of mixed bacterial–fungal suspension for 14 days under aerobic conditions, at a temperature of 35 °C. The suspension was contained with *S. aureus* and *C. albicans*, at a density of 0.5 McFarland. After that, all implants were randomly distributed into four groups, so that there were 20 implants per group (Figure 1). The implants were removed from the test tubes with sterile tweezers. The implants were then put on sterile gauze and, depending on which group they were in, went through the right cleaning process.

### 2.1. Disinfection Protocols

The first group (G1) represented the negative control. This group was not subjected to any disinfection protocol and served as a reference in assessing the effectiveness of a particular disinfection protocol.

The second group (G2) represented the positive control. That group was treated with 0.2% CHX. CHX was applied on a sterile cotton pellet, and the surface of the implant was rubbed with it for 60 s.

The third group (G3) was treated with 3% hydrogen peroxide. H_2_O_2_ was applied on a sterile cotton pellet, and the surface of the implant was rubbed with it for 60 s.

The fourth group (G4) underwent the same procedure as G3, with the following difference: 60 s after treatment with 3% hydrogen peroxide, the samples were illuminated with a SiroLaser Blue laser (Dentsply Sirona, Bensheim, Germany) with the following parameters: wavelength of 445 nm, power of 1 W, continuous beam (CW), 320 mm laser fiber tip and 60 s exposure time (Figure 2). The distance of the laser tip from the implant surface was approximately 1–2 mm, with a constant movement of 1 mm/s along the implant surface (freehand movement).

### 2.2. Collection of Samples

After the processing of the implants, the surface of each of them was scraped with the help of a sterile plastic inoculating loop in such a way, that the threads of the implants were scraped with it twice (Figure 3). Each sample was then placed in 250 µL of saline and vortexed on a vibro-mixer (Corning^®^ LSETM vortex mixer, Corning, NY, USA) for 40 s to separate bacteria and fungi. Then, the loops were discarded, and the physiological solution into which the bacteria and fungi were scraped from the implant were transferred with to the blood agar, in a volume of 50 µL. Then, 100 µL of physiological solution, with suspended *S. aureus* and *C. albicans*, was transferred to the wells of a microtiter plate filled with 100 µL of brain–heart broth.

**Figure 3 biomedicines-11-01002-f003:**
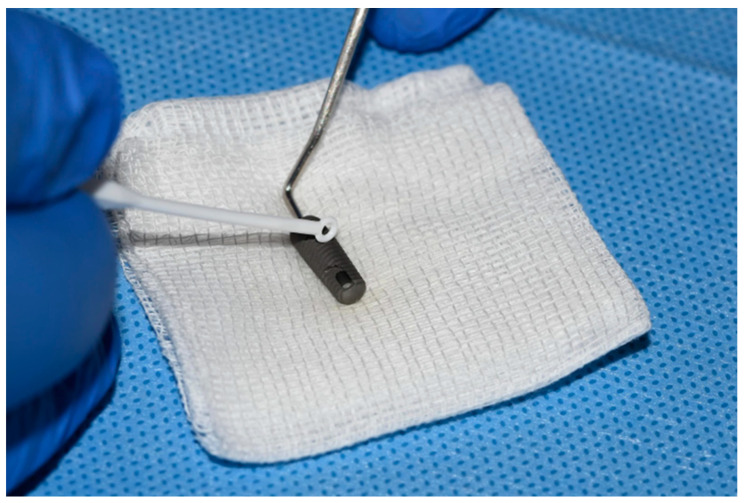
Implant surface sampling with sterile plastic inoculating loop. From the first well, which contains a total of 200 µL (100 µL suspension + 100 µL brain–heart broth), 100 µL was transferred to the next well, and so on, until a dilution of 10^−7^ was reached **(**Figure 4). From each well, 50 µL was inoculated onto blood agar. The plates were then incubated at 35 °C for 48 h. Then, the growth of microbial cultures was read in such a way, that the CFU of individual pathogens was counted, and the number was multiplied by the degree of the dilution (Figure 5).

The formation of bacterial and fungal plaques was confirmed through an SEM device (JSM-7800 F Schottky Field Emission Scanning Electron Microscope, JEOL Ltd., Tokyo, Japan), as seen in Figure 6.

### 2.3. Statistical Analysis

Descriptive statistics of CFU/mL were performed for *S. aureus* and *C. albicans*, depending on the disinfection protocol implemented. Furthermore, a Kruskal–Wallis ANOVA by Ranks analysis was performed, especially for *S. aureus* and especially for *C. albicans*, to prove whether there was a difference between the studied groups. It was followed by post hoc multiple comparison *p*-values (2-tailed), in order to show where there is a statistically significant difference. Then, the protocol was compared individually by the Mann–Whitney U test with continuous correction. Statistical calculations were performed with the TIBCO Data Science Workbench (TIBCO Software, Inc., Palo Alto, CA, USA), software version 14.0.0.15.

## 3. Results

Descriptive statistics of CFU/mL for all protocols for *S. aureus* are shown in Table 1. In the negative control, the highest proportion of colonies per mL was present at 2.86 × 10^8^, followed by the group disinfected with hydrogen peroxide (G3) with 134 CFU/mL, the group of laser-activated hydrogen peroxide (G4) with 45 CFU/mL, and the positive control (G2) with the smallest proportion of 4 CFU/mL. Differences between groups (Mann–Whitney U test) are shown in Table 2 and graphically in Figure 7.

Descriptive statistics of CFU/mL for all protocols regarding *C. albicans* are shown in Table 3. In the negative control, the highest proportion of colonies per mL was present at 4.06 × 10^5^, followed by the group disinfected with hydrogen peroxide (G3) with 24 CFU/mL, the positive control (G2) with 20 CFU/mL, and the laser-activated hydrogen peroxide group (G4) with the smallest proportion of 5 CFU/mL.

Furthermore, there was no statistically significant difference in *C. albicans* eradication between the individually tested protocols: positive control (G2), hydrogen peroxide (G3), and laser-activated hydrogen peroxide (G4) (Table 4 and Figure 8).

## 4. Discussion

This research primarily examined the possibility of improving the antimicrobial efficiency of 3% H_2_O_2_ in peri-implantitis therapy, using diode laser activation with a wavelength of 445 nm, compared to non-activated H_2_O_2_ and the current gold standard of 0.2% CHX [23]. All tested groups showed a massive, significant reduction in microbial CFU count when compared to the negative control. As for the hypothesis, the first hypothesis was only partially accepted. Although the laser-activated group clearly showed better results in the microbial CFU count, there was no statistically significant difference after the statistical analysis, likely due to the sample size. The second hypothesis was accepted, since the analysis revealed comparable results between CHX and H_2_O_2_ groups. H_2_O_2_ is one of the disinfectants used in dental medicine. Oxygen radicals (ROS) generated by the application of H_2_O_2_ have a high affinity for bacteria and the extracellular matrix of bacteria, and a strong oxidizing effect. Additionally, they ensure the creation of active oxygen foam that also has mechanical cleaning properties. ROS (hydroxyl radicals) can theoretically damage healthy tissue cells, but ROS generated by the application of 3% H_2_O_2_ have a rapid breakdown into water and oxygen molecules [24]. Also, the incidence of ROS, which occurs with photoactivation of 3% H_2_O_2_, was examined, and it was shown that the formation of ROS stops when the photoactivation with the laser device stops [25]. In the research of Wiedmer et al. [26], a comparison of the effects of CHX and H_2_O_2_ on the biofilm mass on the implant surface was shown. It has been proven that H_2_O_2_ has a strong effect on actively removing the mass of biofilm, while CHX has a slightly weaker effect on reducing the said mass, but it certainly has a bactericidal effect on the microbial community and prevents the regrowth of bacteria. The current study showed a roughly equal effectiveness of CHX and H_2_O_2_ in eradicating *S. aureus* and *C. albicans*. This study was conducted on the surface of real dental implants rather than titanium discs, in order to simulate in vivo conditions and obtain more realistic results [27,28]. *C. albicans* and *S. aureus* cultures were chosen for research due to the simple cultivation, control of the microorganisms themselves, and also due to the fact that they create polymicrobial biofilms (“early colonizers”) and actively participate in peri-implant infections [13,29].

Diode lasers belong to the newer group of techniques combating the causative agent of periodontitis [30]. They demonstrate a disinfection effectiveness in two main ways: through photothermal and/or photodynamic therapy. The effectiveness of photothermal therapy is based on the thermal energy generated by the laser during radiation emission. The generated thermal energy transfers indirectly through the environment or directly on the bacterial cell, acting fatally on it [31]. Regarding the energy levels of lasers, proper energy dosage is crucial: too little energy will not be sufficient to eradicate enough microorganisms and too much energy could cause damage to the neighboring tissues [32]. Usual antimicrobial energy levels in peri-implant decontamination studies range from 0.5 to 3 W in continuous (CW) or pulsed-mode (energy emission types) [32,33,34,35,36,37]. Photodynamic therapy with diode lasers is based on the light activation of the photosensitive agent in presence of oxygen with a low-energy (sub-ablative) laser beam. There is no heating, and thus no risk of damage to the surrounding tissues [38]. The principles of photodynamic therapy are: photosensitive molecules (the dye) bind to target microorganisms on the implant surface, which is then irradiated with a light of a certain wavelength in the presence of oxygen. Light-excited photosensitizers undergo type I (electron transfer) and/or type II (energy transfer) reactions to produce reactive oxygen species (ROS), resulting in the disruption of the bacterial cell wall and/or normal metabolism, leading to bacterial cell damage or death [38,39]. The described mechanisms do not harm human cells, as these cells have mechanisms to survive oxidative stress (catalase and superoxide dismutase enzymes).

The development of microbial resistance to PDT is not probable, as the bactericidal effect is achieved through the action of oxygen radicals on the cellular components of the micro-organisms [38].

In the current research, a new diode laser, with a wavelength of 445 nm, was used to test the possibility of activating 3% H_2_O_2_. H_2_O_2_ exhibits maximum absorption at wavelengths of approximately 400 nm. The idea was to try to activate H_2_O_2_ photodynamically, and also photothermally (by heating) at the same time; therefore, the energy settings were set to 1 W CW for 60 s, with constant movement of the laser optical fiber (1 mm/s) along the surface of the implant, from a distance of approximately 2 mm. The thermal effect of these settings was not measured, although the settings, combined with the constant displacement/freehand movement of the laser fiber tip in a wet environment (movement prevents excessive heat accumulation on one place and allows cooling microbreaks), could be considered as relatively safe and were therefore chosen for research [32,33,34,35,36]. Nevertheless, additional testing should be done regarding the temperature rise in the specific protocol and its effects on the surrounding tissue.

Chemical reactions in the activated H_2_O_2_ environment were not assessed. Potential activation/antimicrobial augmentation was assessed only via the CFU count, which is a more basic, simpler approach. Complex chemical analysis should be done in future similar studies to closer analyze the interaction of this specific laser wavelength and H_2_O_2_.

The specified wavelength of the diode laser was first examined in the concrete disinfection protocol by Katalinic et al. [40], however not in peri-implant conditions, but on endodontic intracanal biofilms composed of E. faecalis, *C. albicans*, and *S. aureus*, where promising results were obtained. The same laser was then tested in peri-implantitis therapy, but as part of photodynamic therapy with 0.1% riboflavin, where a positive antimicrobial effect was also demonstrated [41]. By reviewing the literature available to the authors, it is not possible to find research similar to the current one. However, research examining the impact of laser energy on H_2_O_2_ and various pathogens exists. In the study by Ikai et al. [42], the authors analyzed the effect of activated hydrogen peroxide on cultures of S. mutans, A. actinomycetemcomitans, E. faecalis, and *S. aureus*. It has been proven that hydrogen peroxide, activated by a laser with a wavelength of 405 nm, has the ability to eliminate all four pathogens within 3 min, which is not the case when using lasers or H_2_O_2_ as independent treatments. Photoactivation of H_2_O_2_, with the aim of improving the antimicrobial effect, was also examined in the work of Mahdi et al. in 2015 [21]. The authors proved a stronger disinfection potential when H_2_O_2_ was activated by LED light with a wavelength of 440–480 nm, compared to non-activated H_2_O_2_. In contrast to the current work, the photoactivation was performed with LED light and not with a diode laser that has monochromatic, coherent, and collimated light radiation. In two scientific papers, Odor et al. [43,44] investigated the antimicrobial effect of hydroxyl radicals produced by diode laser photoactivation, in combination with conventional mechanical periodontitis therapy. A diode laser with a wavelength of 940 nm and a power of 1 W was used. The laser-activated H_2_O_2_ group showed the best results. In relation to the other research mentioned, that research was not done in vitro, but in vivo, on patients with periodontitis, and the effect on bacterial cultures was examined differently from that in the current research.

From all presented studies, it is possible to conclude that laser activation of H_2_O_2_ has a strong, positive and potentially clinically relevant antimicrobial effect, but the results cannot be directly compared with the current study due to too many differences in the design of the study (microorganisms tested, energy settings of the laser, different wavelengths, different presentation of the obtained data, etc.). Additional research is needed to determine the exact impact of 445 nm laser energy on H_2_O_2_ and most efficient laser energy settings, leading to a safe and useful clinical decontamination protocol.

## 5. Conclusions

The conducted research provides a preliminary insight into the protocol for treating the surface of dental implants with a combination of agents that have not been described in the literature so far. Statistical analysis revealed a significant difference between all three disinfection protocols compared to the negative control. However, in the mutual comparison of the results of the three disinfection protocols, there were no statistically significant results, although the laser-activated H_2_O_2_ group showed better antimicrobial results compared to non-activated 3% H_2_O_2_. Within the inherent limitations of this study, it is possible to conclude that all three disinfection protocols are equally powerful in the treatment of S.aureus and C.albicans biofilms. For further evaluation of efficiency, new research is needed on a larger number of implants, testing more energy settings, thermal effects of the settings and, certainly, in-clinical, “in vivo” set-ups.

## Figures and Tables

**Figure 1 biomedicines-11-01002-f001:**
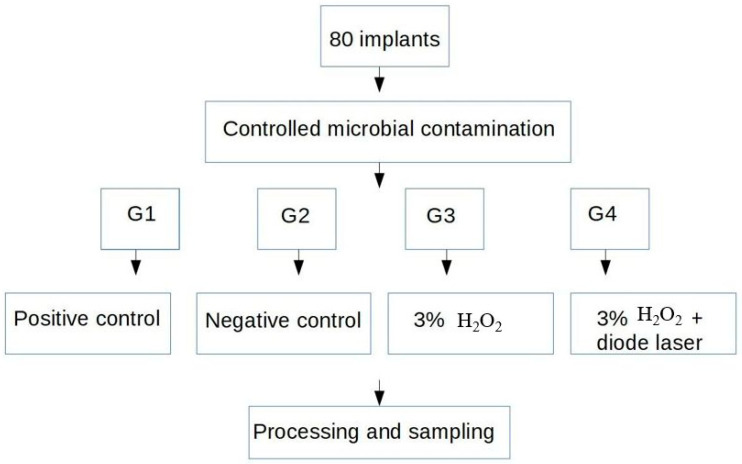
Test sample distribution, contamination and disinfection outline.

**Figure 2 biomedicines-11-01002-f002:**
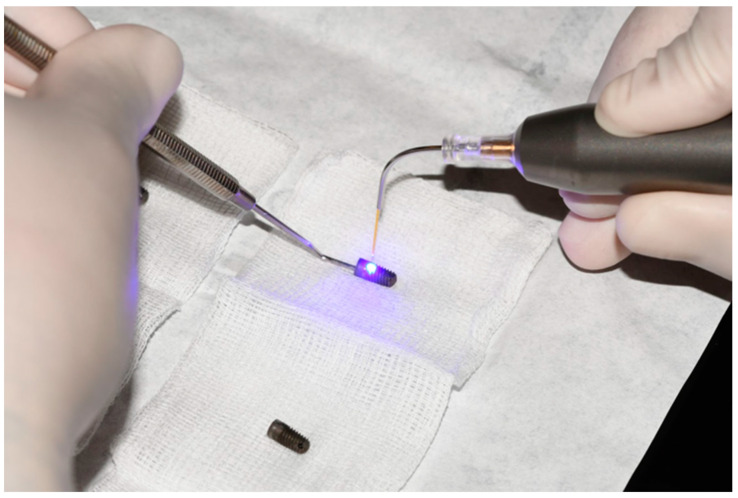
Surface decontamination with SiroLaser Blue laser (Dentsply Sirona, Bensheim, Germany) after treatment with 3% hydrogen peroxide.

**Figure 4 biomedicines-11-01002-f004:**
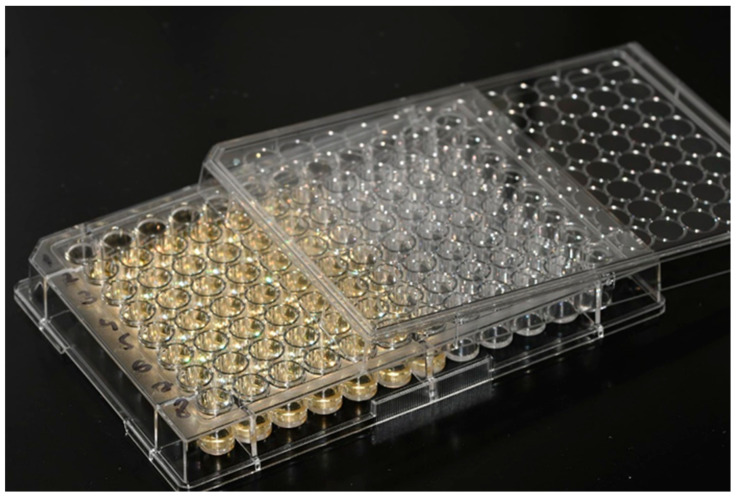
Dilution of samples up to a concentration of 10^−7^.

**Figure 5 biomedicines-11-01002-f005:**
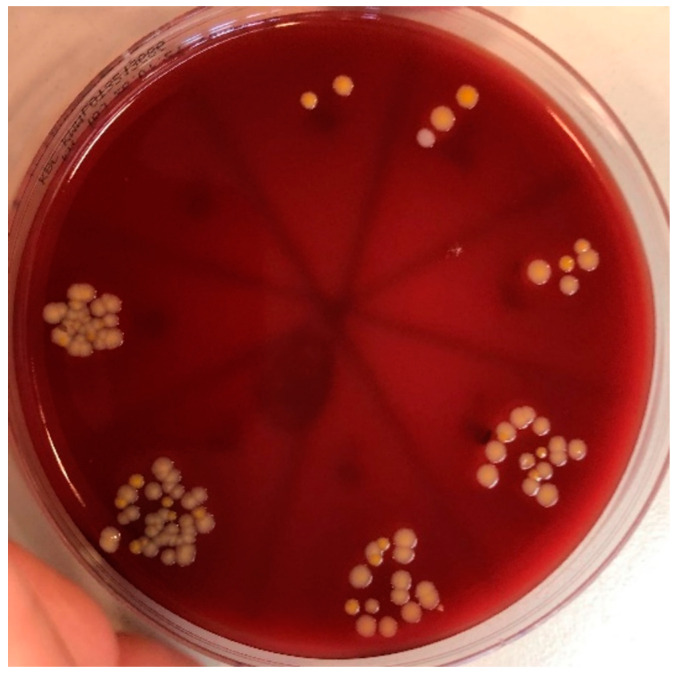
The growth of microbial cultures, ready for CFU/mL counting.

**Figure 6 biomedicines-11-01002-f006:**
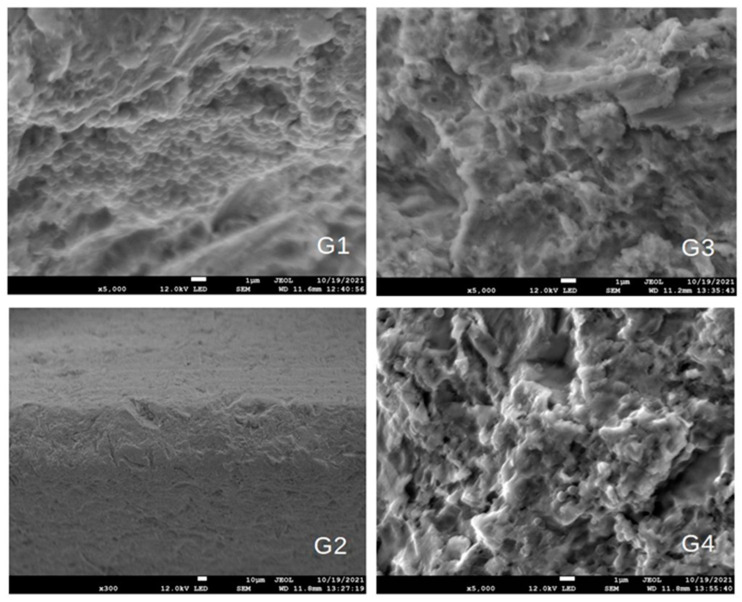
SEM images of G1–G4. Microbial biofilms were visible on the surface of the implant in G1, while they were practically absent in the remaining images.

**Figure 7 biomedicines-11-01002-f007:**
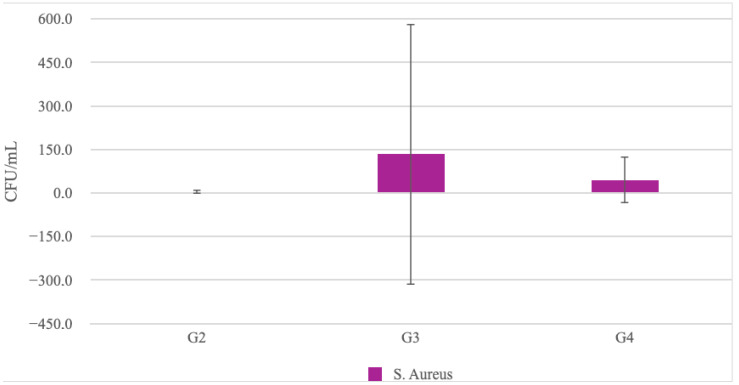
Comparison of the success of individual protocols in the eradication of *S. aureus*: G2—positive control, G3—hydrogen peroxide, and G4—laser-activated hydrogen peroxide.

**Figure 8 biomedicines-11-01002-f008:**
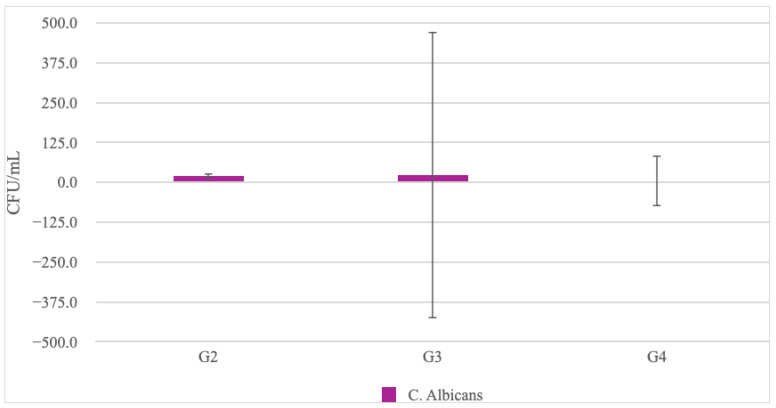
Comparison of the success of individual protocols in the eradication of *C. albicans*: G2—positive control, G3—3% hydrogen peroxide, and G4—laser-activated 3% hydrogen peroxide.

**Table 1 biomedicines-11-01002-t001:** Descriptive statistics of the amount of CFU/mL of *S. aureus* under different protocols.

CFU/mL *S. aureus*	Sample Size	Medium Value	Minimal Value	Maximal Value	Standard Deviation
Negative control	20	286,731,516	20.00	2,000,000,000	508,403,118
Positive control	20	4	0.00	20	8.21
3% H_2_O_2_	20	134	0.00	2000	449.75
Laser + 3% H_2_O_2_	20	45	0.00	200	79.97

CFU/mL—colony-forming units per milliliter.

**Table 2 biomedicines-11-01002-t002:** Comparison of the success of individual protocols in the eradication of *S. aureus*: G1–negative control, G2–positive control, G3–hydrogen peroxide, and G4–laser-activated hydrogen peroxide.

Mann-Whitney U Test By Variable CFU/mL *S. aureus* Marked Tests Are Significant at *p* < 0.05		
**Protocol**	*p*-value	2 × 1 sided exact *p*
**G1 vs. G2**	**0.000000**	**0.000000**
**G2 vs. G3**	0.059363	0.120700
**G2 vs. G4**	0.326615	0.461169
**G3 vs. G4**	0.501578	0.564832

**Table 3 biomedicines-11-01002-t003:** Descriptive statistics of CFU/mL of *C. albicans* under different protocols.

CFU/mL *C. albicans*	Sample Size	Medium Value	Minimal Value	Maximal Value	Standard Deviation
Negative control	20	406,240	0.00	4,000,000	959,808
Positive control	20	20	0.00	200	61.56
3% H_2_O_2_	20	24	0.00	200	60.73
Laser + H_2_O_2_	20	5	0.00	40	11.00

CFU/mL—colony-forming units per milliliter.

**Table 4 biomedicines-11-01002-t004:** A performance comparison of different protocols in the reduction of the *C. albicans* CFU.

Multiple Comparisons *p*-Values (2-Tailed); CFU/mL *C. albicans* Independent (Grouping) Variable: Protocol Kruskal–Wallis Test: H (3, N = 80) = 36.32679 *p* = 0.0000				
Protocol	Negative Control R:63.30	Positive Control R:30.10	3% H_2_O_2_ R:36.00	Laser + H_2_O_2_ R:32.60
Negative control		0.000037	0.001219	0.000177
Positive control	0.000037		1.000000	1.000000
3% H_2_O_2_	0.001219	1.000000		1.000000
Laser + H_2_O_2_	0.000177	1.000000	1.000000	

CFU/mL—colony-forming units per milliliter.

## Data Availability

The data presented in this study are available on request from the corresponding author.

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
