# Peer review of "Evaluation of the Photoactivation Effect of 3% Hydrogen Peroxide in the Disinfection of Dental Implants: In Vitro Study"

_biomedicines, 2023, doi:10.3390/biomedicines11041002_

Round 1

Reviewer 1 Report

Abstract

line 20 ''The antimicrobial activity of the groups was tested''. Please add more information about the methodologies of the study in the abstract section.

Line 49 to 57: ''Therefore, over time, numerous antimicrobial procedures and protocols have been devel- oped. Chlorhexidine digluconate (CHX) has been known for many years as the gold  standard in the treatment of periodontal and peri-implant diseases (13). Hydrogen peroxide (H2O2) has been used in dentistry as a mouthwash to prevent plaque and as an antiseptic after oral surgery for more than 100 years (14). Photo-activation of H2O2 with theaim of improving the antimicrobial effect in periodontitis therapy is a potentially interest ing idea examined in the work of Mahdi et al. in 2015 (15). They found a stronger disinfection potential when H2O2 was activated by LED light with a wavelength of 440480 nm compared to unactivated H2O2 (15). '' The state of the art on different antimicrobial and antibacterial procedures is currently described. However it is limited to few procedures, while there are a a lot of different related procedures. A recent article describing a novel method related to this aim was recently published. I suggest you to add it in the paragraph to improve the state of the art with the most recent articles published on the topic:

Alovisi, M.; Carossa, M.; Mandras, N.; Roana, J.; Costalonga, M.; Cavallo, L.; Pira, E.; Putzu, M.G.; Bosio, D.; Roato, I.; Mussano, F.; Scotti, N. Disinfection and Biocompatibility of Titanium Surfaces Treated with Glycine Powder Airflow and Triple Antibiotic Mixture: An In Vitro Study. Materials 202215, 4850. https://doi.org/10.3390/ma15144850

- Please add the study hypothesis at the end of the introduction section after the aim of the study.

Materials and methods:

- Described in more details the surface characteristic of the tested implants including titanium type and surface roughness (they are factors that can influence the bacterial adhesion and biofilm removal)

- Figure 1: the study design is currently missing

- Line 78-79: ''After that,  the implants were randomly distributed into four groups, so that there are 20 implants in each group (Figure 1.).''  To improve the fluency of the article, please move this sentence at the beginning of the next paragraph ''disinfection protocols''. In addition, add at the beginning of the ''study design'' section how many titanium were selected (if 20 implants per each group = 80 implants, but this information  must be stated as 80 implants were selected...)

- In regard to the set images, I think it would be more appropriate to show them in the results section. In the materials and methods section you can say samples were analyzed using set to assess biofilm formation...and then in the results section showing the images and describe them.

Discussion

- Discuss if the hypothesis were accepted or rejected based on the results of the study.

Author Response

Dear Reviewer 1, thank you for your valuable input and help.

Remark 1: We have added more information about the methodologies of the study in the Abstract section, as proposed.

Remark 2, 3: We have boosted the list of contemporary antimicrobial treatments along with proper citations. Hypothesis was added as well (both in Introduction and Discussion).

Remark 4: Materials and methods; we have slightly rewritten/rearranged this part in order to avoid confusion. Also, we have added more details about the implant surface, as proposed.

Remark 5: We agree that SEM images in some studies like https://doi.org/10.1038/s41598-021-99709-8 suit better under the Results tab. Here we believe current outline somehow better follows the study flow since the SEM was done immediately after the groups testing. There it was important for us to register biofilm formation. Since we have mentioned this in the text right after the testing part, we also needed to place an image “tag” along with the text. This outline was used in similar contemporary 445 nm studies (see below) and if possible, we would like to leave it as it is.
https://doi.org/10.3390/bioengineering9070308
https://doi.org/10.1016/j.pdpdt.2019.04.014 

Remark 6: the text regarding Hypothesis was added to the text (both Introduction and Discussion), as proposed.

Overall we have added more data in Introduction and Discussion sections.
Best regards, the Authors

Reviewer 2 Report

1.Good job, good intuition and adequate study design, there are several limitations on the evaluation of the temperature reached by the implants. This investigation is of vital importance for the feasibility of the work and the in vivo applicability. It is necessary to know the temperature in order to be able to establish exactly whether there are risks for the vitality of the bone and the stability of the implants.

2.196 line- The effectiveness of photothermal therapy is based on the thermal energy generated by the laser during radiation emission. The generated thermal energy acts indirectly through the environment or directly on the bacterial cell, acting fatally on it . Photodynamic therapy with diode lasers is based on the light activation of the photosensitive agent with a low-energy laser beam with the help of oxygen. The interaction between the photosensitive substance and the laser beam, as well as the ability to produce free oxygen  radicals in the described process, determines the antimicrobial effect= explain in a more exhaustive way the  mechanisms of photothermal and photodynamic therapy

3. line 205- H2O2 exhibits maximum absorption at wavelengths of approximately 400 nm.The idea was to activate H2O2 photodynamically but also photothermally (by heating) at the same time; therefore, the energy settings were 1 W CW for 60s, with constant movement of the laser optical fiber (1 mm/s) along the surface of the implant  from a distance of approximately 2 mm. The thermal effect of these settings was not measured, although they, along with the constant displacement of the laser fiber, are considered safe in principle and were therefore chosen for research (26).=A system in which it is not possible to measure the parameter being evaluated cannot be considered valid. The temperature of the implant must be calculated in order to establish its efficacy and safety for the surrounding vital tissues.

4. 249-With the inherent limitations of this study, it is  possible to conclude that all three disinfection protocols are equally antimicrobial in the  treatment of peri-implantitis. For further evaluation of efficiency, research is needed on a  larger number of implants, in more energy settings, an certainly in clinical conditions=This statement cannot be made because the experiment did not concern implants affected by peri-implantitis but the decontamination of voluntarily contaminated implant surfaces.

5.Another important limitation for the in vivo analysis is the difference between the angle of incidence of the laser beam on the surface of the implant treated in vitro and the penetration of the beam from the gingival pocket of the implant into the mouth. The increase in the distance that is recorded descending in the corono-apical direction, the impossibility of penetrating the interproximal spaces, the inadequacy of the radius beyond a certain number of mm and above all the effects on the peri-implant bone of a radius which acts below the guidance of an operator who moves blindly.

Author Response

Dear Reviewer 2,
Thank you for your valuable input and help. 

Remarks 1, 3: 
Since two of your remarks focus on the temperature aspect of the experiment, we will try to clear out our intentions with this specific study. We agree that temperature rise can be a critical detail when discussing diode laser decontamination protocols. Temperature evaluation is a complex task, which can be seen from the studies like:
 https://doi.org/10.1038/s41598-021-99709-8
https://doi.org/10.3390/ma12233934
https://doi.org/10.1563/aaid-joi-D-16-00188 
doi: 10.17219/acem/68943.
doi: 10.1007/s10103-010-0876-8.
Results and recommendations emerging from these studies are not 100% clear and uniform. As for the 445 nm diode laser, this wavelength is not studied enough to make more exact conclusions/guidelines. Although our study design/power settings were partially based on findings of mentioned studies, one must be aware of the facts that we have worked in:
1)    Non – contact mode (2 mm distance)
2)    Constant movement of the fiber tip along the wet implant surface (1mm/sec/60s)
This should be enough to keep the temperature rise as low as possible, below the 47C/60s, as previously stated by many authorities on this subject. Thermal studies, like the ones mentioned above, are mainly “stationary” and cannot really relate to our settings. If you consider all facts mentioned, it becomes clear that a temperature aspect is so complex it should be addressed in a whole new study that would deal with different parameters influencing temperature rise (mentioned in Discussion).
Finally, we would like to point out (this is well emphasized in the text – we have even added more text on the subject in the Discussion area) that the aim of the study was NOT thermal effects on the peri-implant tissues but to evaluate if there was ANY influence of 445 nm laser beam on the 3% H2O2 (possible antimicrobial augmentation effect), which, if proven, could lead to further researches that would be more detailed and consequentially bring benefits for new “in vivo” clinical antimicrobial protocols.

Remark 2: We have added more thorough description of the laser antimicrobial mechanisms, as you proposed, and we augmented the part of our Discussion with temperature rise safety considerations and different energy levels/choice of energy levels.

Remark 4. We have rephrased the sentence to be more clear about the conclusions, as you proposed.

Remark 5: Your statements are true in general, but I think we are all aware that “in vitro” studies, although full of inherent limitations, are usually the first (and necessary) step when testing new protocols and ideas. 

Overall, we have added more data in Introduction and Discussion sections.
Best regards, the Authors

Round 2

Reviewer 1 Report

Dear Authors,

Thank you for addressing my points.

Author Response

Dear Reviewer, once again thank you for your help and guidance. Best regards, the Authors

Reviewer 2 Report

1-Substituite H2O2 with H2O2

2-Line 35:" Peri-implant diseases (peri-implant diseases mucositis and periimplantitis) are probably the most significant issues associated with dental implants (3, 4)". Considering the importance of the chemical component and the morphology of the implant surface with respect to bacterial colonization, it is advisable to update the bibliography with the recent article:
"Memè, L.; Sartini, D.; Pozzi, V.; Emanuelli, M.; Strappa, E.M.; Bittarello, P.; Bambini, F.; Gallusi, G. Epithelial Biological Response to Machined Titanium vs. PVD Zirconium-Coated Titanium: An In Vitro Study. Materials 2022, 15, 7250. https://doi.org/10.3390/ma15207250"

3-Line 244: Authors find reasonable to assume that the tested antimicrobial technique could have similar effects on other organisms usually found in peri-implant infections." It is not correct to speak of the personal opinions of the authors, therefore while sharing the statement it is necessary to modify the text, adhering to the scientific evidence and adding that further in vivo studies will be necessary to validate the therapy studied on the different colonies of microorganisms responsible for peri-implantitis, to do this, please add this reference : Alcohol-free essential oils containing mouthrinse efficacy on three-day supragingival plaque regrowth: " A randomized crossover clinical trial Marchetti, E., Tecco, S., Caterini, E., ...Mattei, A., Marzo, G. Trials, 2017, 18(1), 154."

Author Response

Dear Reviewer, once again thank you for your mentoring and suggestions.

  1. We have substituted H2O2 with H2O2, as proposed
  2. We have read your proposed citation and added it in the Introduction (citation number 15)
  3. You are right about our statement; it is not backed up with evidences nor is it really relevant for the study so we have removed it to avoid further complications.